# Cost-Sensitive Uncertainty-Based Failure Recognition for Object Detection

**Moussa Kassem Sbeyti**[1,2]      **Michelle Karg**[1]      **Christian Wirth**[1]      **Nadja Klein**[3]      **Sahin Albayrak**[2]

[1]Continental AG, Germany
[2]DAI-Labor, Technische Universität Berlin, Germany
[3]Technische Universität Dortmund, Germany

## Abstract

Object detectors in real-world applications often fail to detect objects due to varying factors such as weather conditions and noisy input. Therefore, a process that mitigates false detections is crucial for both safety and accuracy. While uncertainty-based thresholding shows promise, previous works demonstrate an imperfect correlation between uncertainty and detection errors. This hinders ideal thresholding, prompting us to further investigate the correlation and associated cost with different types of uncertainty. We therefore propose a cost-sensitive framework for object detection tailored to user-defined budgets on the two types of errors, missing and false detections. We derive minimum thresholding requirements to prevent performance degradation and define metrics to assess the applicability of uncertainty for failure recognition. Furthermore, we automate and optimize the thresholding process to maximize the failure recognition rate w.r.t. the specified budget. Evaluation on three autonomous driving datasets demonstrates that our approach significantly enhances safety, particularly in challenging scenarios. Leveraging localization aleatoric uncertainty and softmax-based entropy only, our method boosts the failure recognition rate by 36-60% compared to conventional approaches. Code is available at `https://mos-ks.github.io/publications`.

## 1   INTRODUCTION

Although object detectors exhibit high performance on benchmark datasets, their reliability in real-world scenarios can be undermined by factors such as sensory noise and rare events [Li et al., 2022]. Current detectors typically provide the coordinates of objects along a class and a confidence score. They however often demonstrate overconfidence [Gal and Ghahramani, 2016, Zhang et al., 2023] or produce displaced bounding boxes [Harakeh et al., 2020]. Therefore, in safety-critical applications such as autonomous driving, detectors must exhibit low failure rates by refraining from a detection when its reliability is compromised [Zhang et al., 2023]. This can be achieved by equipping the detector with the capability to estimate its own uncertainty.

Two types of uncertainty are typically distinguished. Aleatoric uncertainty captures inherent noise and variability in the data, and epistemic uncertainty reflects the limitations of the model [Kendall and Gal, 2017]. Previous works demonstrate a correlation between both types of uncertainty and detection errors [Le et al., 2018, Miller et al., 2018, Kassem Sbeyti et al., 2023]. The correlation is strengthened by calibrating the uncertainty [Kassem Sbeyti et al., 2023].

Nevertheless, there still is a significant overlap between the class confidences [Corbière et al., 2019] and uncertainties [Kassem Sbeyti et al., 2023] corresponding to correct (CDs) and false (FDs) detections. This overlap results in CDs that turn into missing detections (MDs) when removing detections. It is therefore crucial to quantify the uncertainty in both the localization and classification heads of a detector since each can fail independently [Choi et al., 2021]. In this work, we further analyze the overlap and explore the impact of different uncertainty types and their calibration w.r.t failure recognition.

Determining failures often relies on manual thresholds set on class confidences or uncertainties, lacking a systematic approach, particularly for object detection [Le et al., 2018, Harakeh et al., 2020]. The probabilistic classification output constrains the classification uncertainty within 0 to 1. However, the uncertainty linked to regressive localization, and consequently the combination of both uncertainties, does not adhere to such constraints. This hinders a straightforward selection of the threshold, e.g., by setting an interpretable 75% confidence threshold. We therefore present an automatic method for cost-sensitive thresholding. Our ap-

proach determines the optimal threshold while considering the associated cost of thresholded CDs due to the previously mentioned overlap. This enables an effective management of the risk associated with a detector in safety-critical systems, since the impact of FDs and MDs varies depending on context. For example, when detecting edible mushrooms, it is more crucial to avoid FDs (misclassifying poisonous mushrooms) than MDs (missing edible mushrooms). However, when detecting the authenticity of archeological artifacts, it is more important to minimize MDs (missing valuable artifacts) rather than FDs (mistaking the genuineness of replicas). Given that failures are characterized by both FDs and MDs, and the cost of each is application-specific, our method aims to allow the prioritization of one over the other via a budget, i.e., the desired bound on the portion of one of the two failure sources, hence introducing a different perspective to risk-control in thresholding detections.

The impact of thresholding detections based on uncertainty on the overall performance of the detector remains unclear, since both CDs and FDs may be removed. We therefore formally derive the minimum requirements on the discarded portions of CDs and FDs to safeguard performance and introduce two metrics to measure the effectiveness of the determined threshold. Overall, the contributions of this paper can be summarized as follows.

- We investigate the potential of classification and localization epistemic and aleatoric uncertainties and the effect of their calibration w.r.t. failure recognition under the assumption of an imperfect correlation between uncertainty and false detections.
- We introduce an automated and optimized algorithm for cost-sensitive uncertainty thresholding thereby leading to safer object detectors that can discard detections.
- We define metrics and requirements to analyze the efficacy of uncertainty-based thresholding.

## 2 RELATED WORK

Failure recognition can be broadly classified into three categories. Input-dependent [Zhang et al., 2014, Daftry et al., 2016, Saxena et al., 2017], feature-dependent [Cheng et al., 2018, Rahman et al., 2019, 2021], and output-dependent methods, including uncertainty-based thresholding, [Grimmett et al., 2013, Hendrycks and Gimpel, 2016, Liang et al., 2017, DeVries and Taylor, 2018, Miller et al., 2018, Corbière et al., 2019].

There are two main output-dependent approaches to failure recognition in classification tasks. The first approach relies on class confidences. The intuition is that they can offer valuable information when compared across examples that include misclassifications, despite that individual confidences may not always be reliable indicators of overall confidence.[Hendrycks and Gimpel, 2016, Liang et al.,

2017, DeVries and Taylor, 2018, Corbière et al., 2019]. The second approach utilizes uncertainty [Grimmett et al., 2013, Triebel et al., 2016, Miller et al., 2018]. Geifman and El-Yaniv [2017] explore both in a classifier, revealing varying performance depending on the dataset. They incorporate a reject option by manually setting a limit on the probability of misclassification, at a trade-off in the probability of non-rejection. Part of their future work is translating the concept to object detection, which we address in this paper.

In object detectors, Le et al. [2018], Harakeh et al. [2020] find that the uncertainty of correct detections is lower than that of false detections. Kassem Sbeyti et al. [2023] discover a correlation between aleatoric localization uncertainty and both mislocalizations and misclassifications. However, they also discover an overlap between the uncertainty of correct and false detections. Therefore, the reliability of confidences and uncertainties is crucial for thresholding. The latter can be increased for confidences [Bahnsen et al., 2014, Guo et al., 2017, DeVries and Taylor, 2018, Zhang et al., 2023] and uncertainties [Kuleshov et al., 2018, Laves et al., 2021, Kassem Sbeyti et al., 2023] via their calibration. Kassem Sbeyti et al. [2023] show that normalizing the localization uncertainty by the size of the corresponding bounding box also enhances its reliability, especially for small objects. Hence, we investigate the effect of calibration and normalization on failure recognition.

In summary, existing works demonstrate promising results for the utilization of uncertainty and class confidences for failure recognition. Yet, the challenge of translating the process to object detectors and jointly considering various uncertainty types persists. So far, the threshold is determined using only a single criterion through manual selection relative to performance metrics [DeVries and Taylor, 2018, Le et al., 2018, Harakeh et al., 2020], risk-coverage analysis [Geifman and El-Yaniv, 2017], or by training models on the misclassification cost [Sheng and Ling, 2006]. Unlike our work, these methods also do not consider a risk analysis compatible with object detection. This includes the non-probabilistic cascade architecture [Cai and Vasconcelos, 2018, Rahman et al., 2021]. They do not account for the distinct costs associated with missing and false detections.

## 3 COST-SENSITIVE DETECTION

Object detectors do not consider the different costs of missing and false detections by default, which may be problematic in real-world scenarios. Therefore, we extend the concept of cost-sensitive learning in classification of Thai-Nghe et al. [2010] to the thresholding of the output of a detector in Sec. 3.1 and derive the requirements for thresholding in Sec. 3.2. We describe our automatic thresholding method along with our metrics to measure its effectiveness in Sec. 3.3. Furthermore, we propose an optimization step combining different uncertainty types for enhancing the per-

formance of the failure recognition system in Sec. 3.4. Our approach is implemented during post-processing, making it compatible with *any* pre-trained detector that outputs at least one uncertainty for both classification and localization.

## 3.1 BUDGET AND COST-SENSITIVITY

Consider a detector that predicts an output $y \in \mathcal{D}$ in the detection set $\mathcal{D}$ with a corresponding uncertainty $\sigma \in \mathbb{R}^+$. For each $y$, its $\sigma$ is compared against a predetermined threshold $\delta \in \mathbb{R}^+$ using the thresholding function $u(\sigma, \delta) = I(\sigma > \delta)$, where the indicator function $I$ is one if $\sigma$ exceeds $\delta$ and zero otherwise. This comparison process, known as uncertainty-based thresholding, categorizes the detection set $\mathcal{D}$ based on $u$ into the two subsets

$$\mathrm{CD_T} = \{y \in \mathcal{D} \mid \sigma \leq \delta, u(\sigma, \delta) = 0\}$$
$$\mathrm{FD_T} = \{y \notin \mathcal{D} \mid \sigma > \delta, u(\sigma, \delta) = 1\},$$

such that $\mathrm{CD_T}$ contains detections that are *assigned* as correct and thus retained, whereas $\mathrm{FD_T}$ contains those that are *assigned* as false and removed.

We further define the *true* category of a detection $y$ based on its class ($c_y$) and its intersection over union ($\mathrm{IoU}(y^*, y)$) with the detection ground truth $y^* \in \mathcal{D}$. This yields

$$\mathrm{CD} = \{y \in \mathcal{D} \mid y^* \in \mathcal{D} \mid c_y = c_{y^*} \text{ and } \mathrm{IoU}(y^*, y) \geq \tau\}$$
$$\mathrm{FD} = \{y \in \mathcal{D} \mid y^* \notin \mathcal{D} \mid c_y \neq c_{y^*} \text{ or } \mathrm{IoU}(y^*, y) < \tau\},$$

where a detection is considered correct if both $c_y = c_{y^*}$ and $\mathrm{IoU}(y^*, y) \geq \tau$ and false if either $c_y \neq c_{y^*}$ or $\mathrm{IoU}(y^*, y) < \tau$. Here, $\tau$ is a manually pre-defined IoU threshold. Finally, missing detections, i.e., $y^*$ without a corresponding $y$, and background instances are defined as

$$\mathrm{MD} = \{y \notin \mathcal{D} \mid y^* \in \mathcal{D}\}$$
$$\mathrm{BG} = \{y \notin \mathcal{D} \mid y^* \notin \mathcal{D}\}.$$

Well-trained detectors tend to produce more CDs than FDs. Therefore, cost-indifferent thresholding typically results in a significant loss of CDs compared to FDs. To enable cost-sensitive thresholding, we follow Thai-Nghe et al. [2010] and assume the cost-matrix summarized in Tab. 1. We thereby assume no cost $C$ for correctly re-

Table 1: Cost-matrix for detection thresholding.

|  | CD | FD |
|---|---|---|
| $\mathrm{CD_T}$ | $C_{\mathrm{CD}} \cdot |\mathrm{CD}|$ | $C_{\mathrm{FD}} \cdot |\mathrm{FD}|$ |
| $\mathrm{FD_T}$ | $C_{\mathrm{MD}} \cdot |\mathrm{MD}|$ | $C_{\mathrm{BG}} \cdot |\mathrm{BG}|$ |

tained or discarded detections, such that $C_{\mathrm{CD}} = C_{\mathrm{BG}} = 0$. The total cost of thresholding detections is thus given by $C_{\mathrm{total}} = C_{\mathrm{MD}} \cdot |\mathrm{MD}| + C_{\mathrm{FD}} \cdot |\mathrm{FD}|$, where $|A|$ denotes the

cardinality of a set $A$. Since $C_{\mathrm{MD}}$ and $C_{\mathrm{FD}}$ differ from one application to the other and are challenging to define, we target the minimization of the total cost by controlling the cardinalities $|\mathrm{MD}|$ and $|\mathrm{FD}|$ instead of their corresponding costs.

For that, let $b \in [0, 1]$ be a pre-defined budget and let $i$, $m$ be the proportions post-thresholding of remaining CDs and removed FDs, respectively. Our cost-sensitive strategy allows the control of the decrease in $|\mathrm{FD}|$ by setting $b \cdot |\mathrm{FD}|$ as a lower bound on $m$ on the one hand. On the other hand it allows the control of the decrease in $|\mathrm{CD}|$, i.e., increase in $|\mathrm{MD}|$, by setting $b \cdot |\mathrm{CD}|$ as a lower bound on $i$. The latter is necessary in common cases of overlap between the $\sigma$ of FDs and CDs resulting in a loss of CDs through thresholding.

Pre-selecting $b$ on either error source allows the prioritization and control of the risk associated with a specific type of error. In the context of autonomous driving, safety regulations or backup algorithms in the system may necessitate a $b$ of, for instance, 0.01, i.e., 1% for undetected objects ($|\mathrm{MD}|$). Similarly, when dealing with poisonous mushrooms, the human body might for example only tolerate 5% of poisonous mushrooms mistakenly identified as edible ($|\mathrm{FD}|$). Financial constraints may also influence the choice of $b$, reflecting the capacity to allocate resources for additional verification of FDs or further detection of MDs.

## 3.2 THRESHOLDING REQUIREMENTS

Our objective is to define criteria that directly link the efficacy of thresholding to the detector performance. Rearranging and applying basic algebraic operations to Eq. (1a) below results in the requirements of Eqs. (2a) and (2b), which ensure an uncompromised detection performance post-thresholding. This involves assessing the metrics

**Recall:**
$$\frac{|\mathrm{CD}|}{|\mathrm{CD}| + |\mathrm{MD}|} \leq \frac{i|\mathrm{CD}|}{i|\mathrm{CD}| + |\mathrm{MD}|}$$

**Precision:**
$$\frac{|\mathrm{CD}|}{|\mathrm{CD}| + |\mathrm{FD}|} \leq \frac{i|\mathrm{CD}|}{i|\mathrm{CD}| + (1-m)|\mathrm{FD}|}$$

$$(1a)$$

**F1-Score:**
$$\frac{|\mathrm{CD}|}{|\mathrm{CD}| + 0.5|\mathrm{FD}| + 0.5|\mathrm{MD}|}$$
$$\leq \frac{i|\mathrm{CD}|}{0.5((1+i)|\mathrm{CD}| + (1-m)|\mathrm{FD}| + |\mathrm{MD}|)}$$

for *all detections* (left) vs. *post-thresholding* (right) with the proportions $i$ of remaining CDs and $m$ of removed FDs. The critical point before the detector performance worsens marks the minimum efficacy in failure recognition via thresholding required to improve safety.

$$\begin{cases} 1 - i \leq m & (2a) \\ (1-i)(|\mathrm{FD}| + |\mathrm{CD}| + |\mathrm{MD}|) \leq m|\mathrm{FD}| & (2b) \end{cases}$$

Note that Eq. (1a) shows that recall can only decrease via thresholding, since no additional detections are introduced. Therefore, the proportion of falsely discarded CDs must be as low as possible, i.e, minimize $\{(1-i), 0\}$. The interpretation of Eq. (2a) is that the proportion of falsely discarded CDs must be lower than that of correctly discarded FDs. Eq. (2b) extends the requirement towards the cardinalities and tightens it by including all falsely discarded detections.

## 3.3 THRESHOLDING AUTOMATION AND EVALUATION

The common manual process of selecting $\delta$ in $u(\sigma, \delta)$ is inconsistent and time-consuming. Our cost-sensitive method outlined in Algorithm 1 and Fig. 2 leverages the Receiver Operating Characteristic curve [ROC curve; Fawcett, 2006] to automate it. The ROC curve compares the false positive rate (FPR) to the true positive rate (TPR) across all values of the uncertainty threshold $\delta \in \mathbb{R}^+$. This allows an interpretable selection of the budget $b$ on the proportion of correctly identified CDs or FDs that should be exploited by $\delta$. Thereby, $\delta(b, \tau)$ is defined as the distinct threshold used to calculate an operating point $(\text{FPR}(\delta(b, \tau)), \text{TPR}(\delta(b, \tau)))$ on the ROC curve for a given $b$ and an IoU threshold $\tau$. As mentioned in Sec. 3.1, $\tau$ controls the *true* category (CD or FD) of all $y \in \mathcal{D}$. To generate the ROC curve, we use the ground truth $y^*$ in the validation set in relation to $y$ and the selected $\tau$ and compare the resulting *true* category to the *assigned* one via thresholding ($\text{CD}_\text{T}$ or $\text{FD}_\text{T}$).

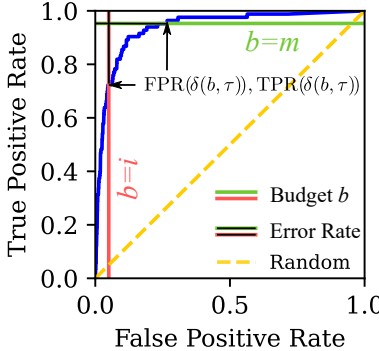

Figure 1: Illustration of the two cost-sensitive use-cases on the ROC curve: Fixing the reduction in CDs and therefore increase in MDs via FPR (red, $1-b = 1-i = 0.05$) or the reduction in FDs via TPR (green, $b = m = 0.95$).

Fig. 1 illustrates the automation of thresholding via the ROC curve for the two use-cases. In the first use-case, the objective is to preserve $i$ CDs bound by $b$ via a maximum FPR (CDs falsely assigned as FDs) while detecting as many FDs as possible. The second use-case prioritizes identifying $m$ FDs bound by $b$ via a minimum TPR (FDs correctly assigned as FDs), regardless of CDs turning into MDs. For the example of $1 - b = 1 - i = 0.05$ (retain 95%

of CDs), the thresholding error is calculated as the corresponding false negative rate (FNR), i.e., 1-TPR. The latter represents the proportion of falsely *assigned* FDs. With $b = m = 0.95$ (remove 95% of FDs), the FPR directly represents the error, i.e., the proportion of falsely *assigned* CDs. Theorem 1 below ensures the determination of the optimal uncertainty threshold $\delta_\text{opt}$ for a pre-defined $b$ and IoU threshold $\tau$. Lemma 1 then states the role of $\delta_\text{opt}$ in the cost-sensitive framework.

**Theorem 1** *The optimal uncertainty threshold* $\delta_{opt}(b, \tau) \in \mathbb{R}^+$ *either maximizes the TPR while the FPR is bound by* $b = i$, *or minimizes the FPR while the TPR is bound by* $b = m$. *It is used to calculate a distinct operating point* $(FPR(\delta(b, \tau)), TPR(\delta(b, \tau)))$ *on the ROC curve given the bugdet $b$ and IoU threshold $\tau$.*

$$\delta_{opt}(b, \tau) = \begin{cases} \arg\max_{\substack{\delta(b,\tau) \in \mathbb{R}^+ \\ FPR(\delta(b,\tau)) \leq 1-b}} TPR(\delta(b, \tau)) & \text{if } b = i \\ \arg\min_{\substack{\delta(b,\tau) \in \mathbb{R}^+ \\ TPR(\delta(b,\tau)) \geq b}} FPR(\delta(b, \tau)) & \text{if } b = m \end{cases} \quad (3)$$

**Lemma 1** *For a given budget $b$, the ROC curve guarantees the existence of an optimal threshold $\delta_{opt}(b, \tau) \in \mathbb{R}^+$ that provides a tailored solution for the specified budget constraints, therefore controlling the recognition error between CDs (bounding the FPR) and FDs (bounding the TPR).*

To evaluate the thresholding performance w.r.t. the detector, we extract the two metrics

$$\text{CD@FD}(b) = \sum_{\tau=0.5}^{0.75} \text{TNR}(\text{TPR}(\delta(b, \tau)))$$
$$\text{FD@CD}(b) = \sum_{\tau=0.5}^{0.75} \text{TPR}(\text{FPR}(\delta(b, \tau)))$$
$$(4)$$

from the ROC curve, where TNR is the true negative rate, i.e., 1-FPR. Hence, $\text{CD@FD}(b)$ denotes the correctly identified CDs at a fixed portion $b$ of correctly identified FDs, while $\text{FD@CD}(b)$ represents the correctly identified FDs at a fixed portion $b$ of correctly identified CDs. Both are calculated for IoU threshold $\tau \in [0.5, 0.75]$, as per usual object detection practices, with a 0.05 step. These cost-sensitive IoU-relative metrics are necessary to quantify the thresholding effectiveness in the context of object detection.

## 3.4 THRESHOLDING OPTIMIZATION

Combining epistemic and aleatoric classification and localization uncertainties, $\sigma_\text{cls}$ and $\sigma_\text{loc}$, has not yet been investigated for failure recognition. Algorithm 1 outlines our approach for an optimized combination via a weighted sum of the uncertainties $\mathbf{w}^\top \times (\sigma_\text{cls}, \sigma_\text{loc})^\top$, $\mathbf{w} = (w_1, w_2)^\top \in [0, 1]^2$ aiming at maximizing $\text{CD@FD}(b)$ or $\text{FD@CD}(b)$

depending on the selected use-case and budget $b$. As summarized in Theorem 2, the optimization process aims to find the combination of weights that results in the most effective $\delta_{opt}$ with the smallest overlap between CDs and FDs.

**Theorem 2** *Let $\Theta = [0,1]^2$ be the optimization search space for the weights $\mathbf{w}$ in the weighted sum of the uncertainties $\mathbf{w}^\top \times (\sigma_{cls}, \sigma_{loc})^\top$. For a given budget $b$, the uncertainty threshold $\delta_{opt}(b, \tau)$ on the weighted sum is derived from the ROC curve along the corresponding FNR and FPR as per Sec. 3.3. These comprise the loss per step $\mathcal{L}_{step}$. The optimization loss is*

$$\mathcal{L}_{opt} = \frac{1}{6} \sum_{\tau \in \mathcal{T}} \begin{cases} \mathcal{L}_{step} = FNR(\delta_{opt}(b, \tau)) & \text{if } b = i \\ \mathcal{L}_{step} = FPR(\delta_{opt}(b, \tau)) & \text{if } b = m \end{cases}$$

*with $\mathcal{T} = \{0.5, 0.55, 0.6, 0.65, 0.7, 0.75\}$. Then, any black-box parameter optimizer converges to $\mathbf{w}_{opt} \in \Theta$ that minimizes $\mathcal{L}_{opt}$. Given that FPR=1-TNR and FNR=1-TPR, minimizing $\mathcal{L}_{opt}$ maximizes the TPR or TNR depending on $b$ and the use-case. Thus, minimizing $\mathcal{L}_{opt}$ also maximizes the thresholding metrics CD@FD(b) or FD@CD(b) in Eq. (4).*

**Remark 1** *As per Sec. 2, both $\sigma_{cls}$ and $\sigma_{loc}$ are crucial for representing the two heads of the detector. Note that $\sigma_{cls}$ may be replaced by the entropy $\sigma_{ent} = -\sum_{l=1}^c p_l \log_2 p_l$ over the confidences $p_l$ of the $c$ classes. We also reiterate the importance of calibrating $\sigma$, denoted by $\sigma_{cls,c}$, and additionally normalizing $\sigma_{loc}$ by dividing it by the bounding box dimensions, denoted by $\sigma_{loc,c,n}$.*

Our approach is illustrated in Fig. 2. In summary, the default output of a detector from stage I undergoes a post-processing step consisting of thresholding. The optimal uncertainty threshold $\delta_{opt} \in \mathbb{R}^+$, along with the weights $\mathbf{w}_{opt} \in [0,1]^2$ for combining the classification and localization uncertainties $\sigma_{cls}$ and $\sigma_{loc}$, and the thresholding metrics for evaluation are all extracted in stage II. The process is constrained by a pre-defined budget $b = i$ for remaining CDs or $b = m$ for removed FDs depending on the application. Stage III illustrates the reallocation of the detections post-thresholding. FDs are successfully discarded, i.e., they become BG, in exchange for a potential loss of CDs that turn into MDs.

# 4 EXPERIMENTS

**Implementation Details.** We select the state-of-the-art detector EfficientDet-D0 [Tan et al., 2020, Google, 2020] pretrained on COCO [Lin et al., 2014] as the baseline and fine-tune it on two commonly used autonomous driving datasets: KITTI [Geiger et al., 2012] with all 7 classes and a 20% split for validation, and BDD100K [Yu et al., 2020] with all 10 classes and the 12.5% official split, for 500 epochs with 8 batches each and an input image resolution of

---

**Algorithm 1** Outline of our approach for an automated and optimized failure recognition process in object detectors.

**Require:**
    $y, y^*, \sigma_{cls}, \sigma_{loc}$     ▷ Detections, labels and uncertainties
    $b = i$ or $m$                       ▷ Budget
1: $\sigma_{cls} = \sigma_{cls,c}, \sigma_{loc} = \sigma_{loc,c,n}$    ▷ Calibrate and normalize
2: $\mathbf{w} \in [0,1]^2$                 ▷ Define search space
3: **for** $i \leftarrow 0$ **to** 50 **step** 1 **do**        ▷ Start optimization
4:      **for** $\tau \leftarrow 0.5$ **to** 0.75 **step** 0.05 **do**
5:          Define CDs, FDs for $\tau$ based on $y^* \leftrightarrow y$
6:          ROC $\leftarrow \mathbf{w}^\top \times (\sigma_{cls}, \sigma_{loc})^\top$, CDs, FDs
7:          **if** $b = i$ **then**
8:              $\mathcal{L}_{step} \leftarrow$ FNR $\leftarrow$ TPR $\leftarrow$ FPR $\approx$ 1-i
9:          **else**
10:             $\mathcal{L}_{step} \leftarrow$ FPR $\leftarrow$ TPR $\approx$ m
11:          **end if**
12:      **end for**
13:      $\mathcal{L}_{opt} \leftarrow \sum_{\tau=0.5}^{0.75} \mathcal{L}_{step}$
14: **end for**                      ▷ End optimization
15: $\mathbf{w}_{opt} \leftarrow \arg\min_{\mathbf{w}} \mathcal{L}_{opt}$      ▷ Optimal weights
16: $\delta_{opt} \leftarrow$ ROC            ▷ Optimal threshold
17: **if** $b = i$ **then**                     ▷ Output
18:      Return $\mathbf{w}_{opt}, \delta_{opt}$, FD@CD(b)
19: **else**
20:      Return $\mathbf{w}_{opt}, \delta_{opt}$, CD@FD(b)
21: **end if**

---

1024×512 pixels. All other hyperparameters maintain their default values [Tan et al., 2020]. Moreover, we validate the BDD fine-tuned models on the corner case dataset CODA [Li et al., 2022] on the 8 classes in common to test our method under domain shift.

**Uncertainty Quantification.** We implement 2D spatial Monte Carlo (MC) dropout [Tompson et al., 2015] to estimate the epistemic classification ($\sigma_{ep,cls}$) and localization ($\sigma_{ep,loc}$) uncertainties with a dropout rate of 0.05 with 10 MC samples based on best performance [a rate of 0.1 drastically reduced it, also cf. Stoycheva, 2021]. To estimate the aleatoric uncertainty ($\sigma_{al}$), we apply loss attenuation [LA; Kendall and Gal, 2017] in the localization head only, as it already covers the aleatoric uncertainty per data sample [Kassem Sbeyti et al., 2023]. We denote the uncertainty stemming from a model with LA only with a subscript $_{la}$, while all the other uncertainty types are from a model with MC+LA. We extract and apply softmax on the predicted classification logits to calculate the entropy $\sigma_{ent}$. We employ isotonic regression per-class to calibrate $\sigma_{cls}$ and per-class and per-coordinate for $\sigma_{loc}$ as per Kassem Sbeyti et al. [2023], denoted with a subscript $_c$. The normalized localization uncertainty is $\sigma_{loc,n} = \frac{\sigma_{loc}}{\text{width or height}}$ depending if it corresponds to a $y$- or $x$-coordinate. The localization and classification uncertainties per object are defined as $\sigma_{loc} = \frac{1}{4} \sum_{i=1}^4 \sigma_{loc,i}$ and $\sigma_{cls} = \max(\sigma_{cls,l})$ for $l \in [1, c]$ with $c$ classes, respectively. We use the heteroscedastic

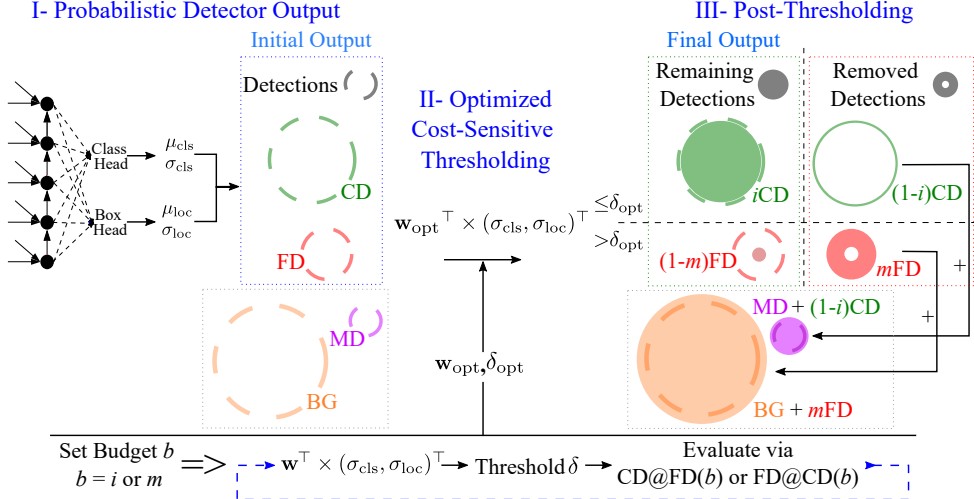

Figure 2: Failure case recognition process via cost-sensitive automated and optimized uncertainty-based thresholding. Circle size symbolizes the typical occurrence rate in well-trained detectors. Dashed circles indicate original detections, solid circles represent remaining detections, and donut-shaped circles signify removed detections (consider grey circles as the legend).

evolutionary Bayesian optimization (HEBO) algorithm of Cowen-Rivers et al. [2022] for the optimization in Sec. 3.4 due to its rapid convergence and ease of implementation. The following results reflect the mean and standard deviation of 3 iterations due to low variation.

**Evaluation Metrics.** Models are evaluated based on the COCO-style average precision [AP; Lin et al., 2014], classification accuracy (Acc), expected calibration error (ECE) of the confidences, and mean IoU (mIoU). The localization uncertainty $\sigma_{\text{loc}}$ is assessed via the negative log-likelihood (NLL). Our approach is evaluated based on the average over $\tau \in \mathcal{T}$ of the Jensen-Shannon divergence [JSD; Lin, 1991], the area under the ROC curve (AUC), our two metrics in Eq. (4) and the balanced accuracy [BAcc; Brodersen et al., 2010], which is equivalent to one minus the uncertainty error [Miller et al., 2019].

## 4.1 UNCERTAINTY ESTIMATION METHODS

We first analyze the effect of implementing off-the-shelf uncertainty estimation methods in a detector. The average inference time per image of the sampling-based method MC dropout measured across the three validation sets on an RTX A5000 is five-fold (190 milliseconds (ms)) that of the baseline (35 ms). The detector with LA is slightly faster (32 ms) due to the Tensor Cores utilization by the extended eight outputs [Appleyard and Yokim, 2017].

Tab. 2 shows that LA also enhances performance on KITTI, while MC dropout decreases it. However, MC dropout performs best on BDD and CODA. This contrast can be attributed to the inherent characteristics of each dataset. MC dropout helps the model handle the noise and diversity via multiple stochastic predictions in BDD/CODA, which in-

clude various weather and time-of-day conditions. In contrast, the additional randomness on the dataset with higher quality KITTI hinders performance, as it introduces unnecessary variability. Meanwhile, LA allows the model output to capture $\sigma_{\text{al}}$, helping it identify the few noisy instances hindering performance.

The localization performance (measured by mIoU) is particularly affected by this trend, while the classification performance (measured by Acc) remains mostly constant. The ECE of the predicted confidences $p$ increases with the adoption of more uncertainty estimation methods due to the increased training complexity and variability. On KITTI and BDD, the quality of $\sigma_{\text{ep,loc}}$ and $\sigma_{\text{al,loc}}$ measured by the NLL is higher when the uncertainty methods are implemented separately. This opposite is true for CODA, which further confirms that inducing more noise into the model is beneficial only when the dataset requires it.

Table 2: KITTI (top), BDD (mid), CODA (bottom): Performance comparison with EfficientDet-D0 baseline.

| Method | AP↑ | Acc↑ | mIoU↑ | ECE $p$↓ | NLL $\sigma_{\text{ep,loc}}$↓ | NLL $\sigma_{\text{al,loc}}$↓ |
|---|---|---|---|---|---|---|
| Baseline | 72.83±0.12 | **0.99±0.00** | 90.06±0.05 | **0.02±0.00** | - | - |
| LA | **73.26±0.50** | **0.99±0.00** | **90.34±0.03** | **0.02±0.00** | - | 3.22±0.01 |
| MC | 70.88±0.17 | **0.99±0.00** | 89.10±0.02 | 0.03±0.00 | **3.09±0.11** | - |
| MC+LA | 70.15±0.09 | **0.99±0.00** | 89.03±0.05 | 0.03±0.00 | 3.17±0.16 | 3.56±0.00 |
| Baseline | 24.69±0.09 | **0.94±0.00** | 67.74±0.07 | **0.12±0.00** | - | - |
| LA | 24.38±0.12 | **0.94±0.00** | 67.69±0.05 | 0.14±0.00 | - | 3.69±0.01 |
| MC | **25.55±0.02** | **0.94±0.00** | 67.30±0.02 | 0.15±0.00 | 26.91±1.73 | - |
| MC+LA | 24.78±0.01 | 0.93±0.00 | 66.60±0.02 | 0.16±0.00 | 22.39±0.77 | 3.78±0.01 |
| Baseline | 16.09±0.07 | **0.89±0.00** | 72.23±0.03 | **0.06±0.00** | - | - |
| LA | 15.53±0.25 | **0.89±0.00** | 72.06±0.14 | 0.08±0.00 | - | 4.27±0.02 |
| MC | **16.97±0.04** | **0.89±0.00** | **73.30±0.08** | 0.11±0.00 | 44.34±3.77 | - |
| MC+LA | 16.05±0.25 | **0.89±0.00** | 72.19±0.03 | 0.12±0.00 | **36.62±3.21** | **4.15±0.03** |

## 4.2 UNCERTAINTY-BASED THRESHOLDING

After examining the cost of estimating $\sigma$, we analyze its usability in recognizing failure cases. Based on Sec. 2, both epistemic and aleatoric $\sigma_{\mathrm{cls}}$ and $\sigma_{\mathrm{loc}}$ are expected to be relevant for the failure recognition. We therefore select for further analysis the combination MC+LA alongside LA only (subscript $_{\mathrm{la}}$) with $\sigma_{\mathrm{ent}}$ as an alternative to $\sigma_{\mathrm{cls}}$ due to the computational costs of MC.

Despite the correlation between multiple $\sigma$ types and failure cases, as discussed in Sec. 2, there exists a substantial overlap between $\sigma$ of CDs and FDs (see the left of Fig. 3). The average $\sigma$ ($\mu_\sigma$) of CDs and FDs does not provide a clear indication of which $\sigma$ type is optimal. We therefore consider the JSD and the AUC. Predicting on the validation set yields a ratio of 2% of FDs over CDs on KITTI and 31% on BDD. Particularly in such scenarios with imbalanced classes (FDs vs. CDs), Fig. 3 (right) highlights the advantage of JSD over AUC, as it captures distributional disparities between the $\sigma$ distributions of CDs and FDs.

Fig. 3 (right) also demonstrates the importance of calibration for $\sigma$, particularly $\sigma_{\mathrm{ep,cls}}$. However, normalizing $\sigma_{\mathrm{loc}}$ by dividing it with the corresponding width and height of its bounding box enhances failure recognition rates the most. Consistent with prior work [Harakeh et al., 2020], $\sigma_{\mathrm{al}}$ proves to be a more discriminative uncertainty estimate for localization compared to $\sigma_{\mathrm{ep}}$, especially in noisy datasets. $\sigma_{\mathrm{ent}}$ and $\sigma_{\mathrm{loc,n}}$ perform best as separation candidates on both datasets. $\sigma_{\mathrm{la}}$ show comparable performance, with a deviation below 1%, to $\sigma$ estimated using MC+LA. This suggests that the performance of non-epistemic uncertainties remains largely unaffected by dropout.

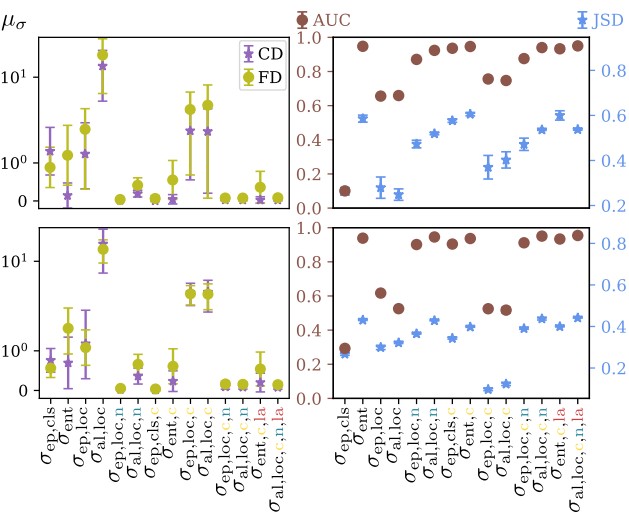

Figure 3: KITTI (top) and BDD (bottom): Comparison between the separation ability of $\sigma$ types based on JSD and AUC (right). On the left is $\mu_\sigma \pm \sigma_\sigma$ of CDs and FDs.

After comparing the separation capabilities of $\sigma$, we selec-

tively retain the eight most promising candidates and discard the rest. The results of thresholding with an exemplary $b = 0.95$ of CDs are presented in Fig. 4 to further analyze the behavior of each $\sigma$ type. We observe that the defined criteria in Eqs. (2a) and (2b) are met at an IoU threshold $\tau$ of 0.8 for KITTI, but instead below 0.5 for BDD. This discrepancy can be attributed to the higher prevalence of FDs in BDD (see count in Fig. 4), making the exclusion of detections based on $\sigma$ a valuable approach.

We notice that $\sigma_{\mathrm{al}}$ of an object detector with LA outperforms its MC+LA variant on both datasets, whereas $\sigma_{\mathrm{ent}}$ does not. This can be attributed to the positive impact of LA on the quality of $\sigma_{\mathrm{al,loc}}$ (see NLL in Tab. 2). Furthermore, Fig. 4 highlights the advantage of $\sigma_{\mathrm{loc}}$ over $\sigma_{\mathrm{cls}}$ the higher $\tau$, since $\sigma_{\mathrm{loc}}$ becomes more indicative of misdetections with stricter IoU requirements due to the independence of the classification error of $\tau$. The performance of $\sigma_{\mathrm{cls}}$s consistently decreases on KITTI in contrast to $\sigma_{\mathrm{loc}}$s. Given the high performance of the object detector on KITTI and the few FDs, $\sigma$ is more tailored to challenging cases in the dataset. As a result, it does not correlate with detections having relatively high IoU but still below $\tau$, reducing the filtering efficiency as $\tau$ increases. As for calibration, it results in a recognition boost only up to a certain $\tau$. This can be traced back to CDs with a lower IoU initially used for calibration now labeled as FDs based on the validation set, hence reducing the separation space between the $\sigma$ of CDs and FDs.

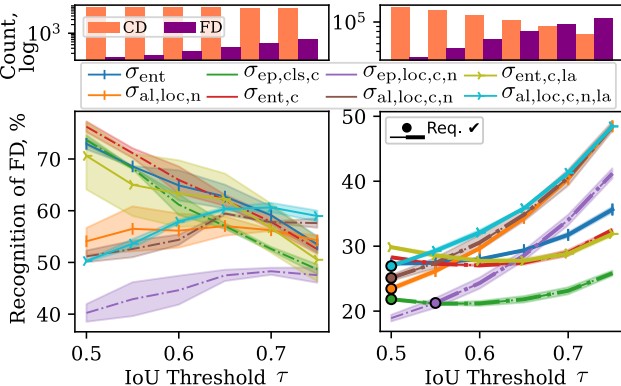

Figure 4: KITTI (left) and BDD (right): Recognition rates of FDs for a fixed budget of 95% CDs. The circles and thicker lines indicate requirement fulfillment in Eqs. (2a) and (2b).

## 4.3 BUDGET BEHAVIOR ANALYSIS

To assess the impact of the budget $b$, Fig. 5 illustrates the introduced metrics in Eq. (4) for $b$ ranging from 0.5 to 0.99. Notably, increasing $b$ of CDs from 0.95 to 0.98 leads to an approximate 50% reduction in detected FDs across all datasets. However, increasing the detection of FDs by 25% results in only a 5–15% decrease in CDs. Recognizing the majority class (CDs) does not incur a significant error due

to their abundance. However, when the focus shifts towards recognizing detections in the overlap region, the error begins to rise steeply. This emphasizes the challenges associated with accurately detecting instances that lie in the ambiguous overlap area illustrated in Fig. 3. $\sigma_{al,loc}$ plays a more significant role on BDD due to the lower localization performance, whereas on KITTI, $\sigma_{ent}$ dominates. The transferability of classification calibration models trained on BDD to CODA reveals that the difference in class characteristics between the two affects the effectiveness of $\sigma_{ent,c}$, as $\sigma_{ent}$ performs best on CODA due to the lower classification performance (see Tab. 2). Overall, the suitability of the different $\sigma$ types does not depend on the budget $b$, but instead on the dataset and the challenges the model still faces at the end of training. While $b$ influences performance as expected, all $\sigma$ types maintain it relatively to each other irrespective of $b$.

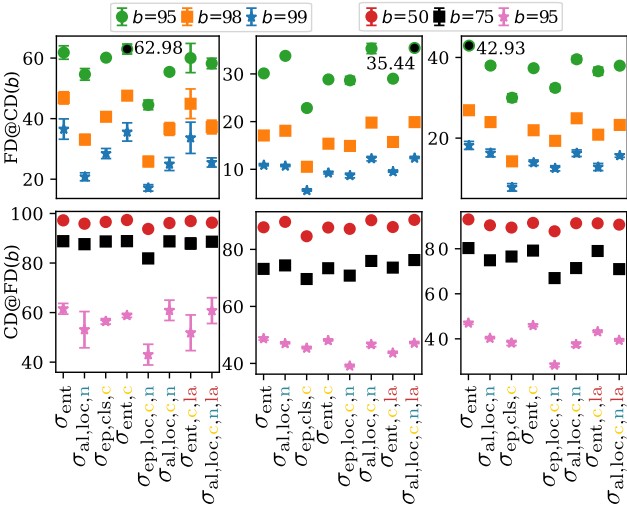

Figure 5: KITTI (left), BDD (mid), CODA (right): Budget effect on the thresholding performance of the $\sigma$ types for different $b$ (%) in both use-cases. Maximum FD@CD($b$) is accentuated for comparison.

## 4.4 OPTIMIZED COMBINED THRESHOLDING

Given our results, we select $\sigma_{ent}$ to represent $\sigma_{cls}$ and $\sigma_{al,loc,c,n}$ to represent $\sigma_{loc}$ and $\sigma_{al}$. We include $\sigma_{ep,loc,c,n}$ to continue the analysis on the usability of $\sigma_{ep,loc}$. We investigate the sum ($\sum$) and its optimization ($\sum *$) of the selected $\sigma$ types. We also explore their combination using multiplication and observe that the sum outperforms it. Tab. 3 demonstrates the benefits of optimization for the example of $b = 0.95$ of CDs. It yields a 3–12% increase in FD@CD95 compared to using separate $\sigma$ types (refer to Fig. 5) or combining them without optimization. $\sigma$ without calibration or normalization results in only up to 15% FD@CD95 on KITTI and 2% on BDD.

Calibrating $\sigma_{ent}$ improves recognition rates on KITTI but not BDD/CODA due to the majority of detections used for

calibration despite lower IoU thresholds, as also described in Sec. 4.2. However, whether using $\sigma_{ent}$ or $\sigma_{ent,c}$, the increase in FD@CD95 of $\sum *$ over $\sum$ falls within similar ranges. $\sigma_{la}$ estimated via LA only and $\sigma_{mc+la}$ estimated via MC+LA also perform similarly ($< 2\%$). Furthermoe, we observe that BAcc is not sufficiently descriptive, hence motivating the usage of our cost-sensitive evaluation metrics. For instance, BDD and CODA exhibit similar BAcc values despite notable differences in recognition rates. Nevertheless, optimization does result in an increase of up to 4% in BAcc. The only cost incurred by the optimization process is the 1–3 minutes average optimization time.

For further analysis, we optimize at both $\tau$s 0.5 and 0.75 separately and notice that $\sigma_{ent}$ plays a smaller role (up to 8%) at higher $\tau$s, while it contributes significantly more (up to a 20%) at lower $\tau$s. $\sigma_{al}$ carries nearly equal importance at both $\tau$s. This behavior aligns with the outcomes depicted in Fig. 4. Furthermore, $\sigma_{ep,loc}$ is deemed redundant and assigned a weight of 0, as we assume that FDs are primarily caused by noisy images rather than a lack of images.

Table 3: KITTI (top), BDD (mid), CODA (bottom): Standard ($\sum$) vs. optimized ($\sum *$) combination of the calibrated and normalized uncertainties for MC+LA and LA only.

| | Weights | | | FD@CD95↑ | BAcc↑ |
|---|---|---|---|---|---|
| | $\sigma_{ent}$ | $\sigma_{ep,loc}$ | $\sigma_{al}$ | | |
| $\sum \sigma_{mc+la}$ | 1.00±0.00 | 1.00±0.00 | 1.00±0.00 | 68.02±1.97 | 0.81±0.01 |
| $\sum * \sigma_{mc+la}$ | 0.16±0.03 | 0.03±0.04 | 1.0±0.00 | **72.36±2.72** | **0.83±0.01** |
| $\sum \sigma_{la}$ | 1.00±0.00 | - | 1.00±0.00 | 65.86±3.43 | 0.80±0.02 |
| $\sum * \sigma_{la}$ | 0.14±0.06 | - | 0.72±0.21 | **70.93±1.47** | **0.83±0.01** |
| $\sum \sigma_{mc+la}$ | 1.00±0.00 | 1.00±0.00 | 1.00±0.00 | 32.03±0.24 | 0.63±0.00 |
| $\sum * \sigma_{mc+la}$ | 0.06±0.03 | 0.00±0.00 | 0.72±0.32 | **37.98±0.90** | **0.67±0.00** |
| $\sum \sigma_{la}$ | 1.00±0.00 | - | 1.00±0.00 | 30.65±0.23 | 0.63±0.00 |
| $\sum * \sigma_{la}$ | 0.05±0.02 | - | 0.72±0.36 | **38.11±0.21** | **0.67±0.00** |
| $\sum \sigma_{mc+la}$ | 1.00±0.00 | 1.00±0.00 | 1.00±0.00 | 40.60±0.21 | 0.68±0.00 |
| $\sum * \sigma_{mc+la}$ | 0.07±0.02 | 0.00±0.00 | 0.82±0.25 | **45.68±0.53** | **0.70±0.00** |
| $\sum \sigma_{la}$ | 1.00±0.00 | - | 1.00±0.00 | 38.49±0.96 | 0.67±0.00 |
| $\sum * \sigma_{la}$ | 0.10±0.01 | - | 0.99±0.00 | **43.95±0.43** | **0.69±0.00** |

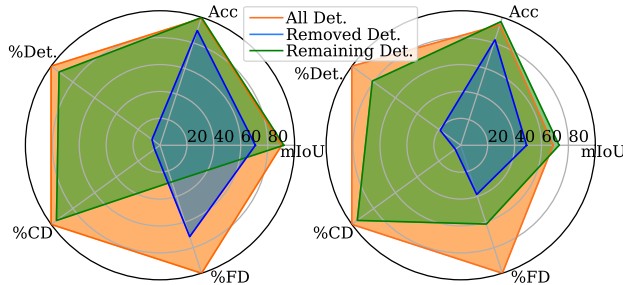

Figure 6: KITTI (left), BDD (right): Effect of thresholding on the classification and localization performance of the detector via Acc, mIoU, and % of removed detections (Det.) including the % of removed CDs and FDs out of the total detections. Values are averaged over $\tau \in \mathcal{T}$.

Discarding detections based on their uncertainty boosts the

mIoU on average by up to 5% and Acc by 1.3% on BDD, with 18% removed detections (incl. 38% FDs and 5% CDs). On KITTI, discarding 7% (incl. 72% FDs and 4.95% CDs) increases the mIoU by up to 2% and the Acc by 0.7%. These findings visualized in Fig. 6 highlight the advantages of detecting more FDs (see count in Fig. 4), while emphasizing the necessity for a risk-aware evaluation. Overall, Fig. 6 illustrates the improvement in both the safety and performance of the detector via the substantial reduction of FDs via our approach while accepting an exemplary pre-defined budget of 5% loss of CDs.

## 5 CONCLUSION

We introduce a cost-sensitive, uncertainty-based, and optimized thresholding approach for failure recognition in object detection, allowing the detector to filter out its false detections during post-processing. We outline the requirements for effective thresholding, propose performance metrics for its evaluation, and investigate the challenges of uncertainty-based thresholding, including the role of epistemic and aleatoric uncertainties and their calibration. We find that a combination of softmax-based entropy and aleatoric uncertainty is optimal, hence avoiding epistemic uncertainty estimation methods and their computational drawbacks. Incorporating LA in a detector also reduces its inference time and enhances its performance. Our approach can remove 38–75% of FDs, at an exemplary cost of up to 5% increase in MDs, which is 3–12% more FDs compared to using separate calibrated and normalized uncertainties and 36–60% to using conventional methods, as in separate unprocessed uncertainties. We hope for this work to redirect the focus in object detection beyond performance to also include considerations of safety, without compromising either aspect.

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
