# OpenReview forum: "Cost-Sensitive Uncertainty-Based Failure Recognition for Object Detection"
_auai.org/UAI/2024/Conference — UAI 2024 oral_

### Official Review · Reviewer_T42T · 2024-03-10

**Q2-1 Originality-Novelty:** 3
**Q2-2 Correctness-Technical Quality:** 3
**Q2-5 Clarity Of Writing:** 3

**Q1 Summary And Contributions:**

This paper aims to develop a post-processing step that allows the detector to filter out its false detections. It introduces a cost-sensitive, uncertainty-based, and optimized thresholding approach for failure recognition. Specifically, it derives minimum thresholding requirements to prevent performance degradation and defines metrics to assess the applicability of uncertainty for failure recognition. Then, it proposes to automate and optimize the thresholding process to maximize the failure recognition rate w.r.t. the specified budget. The results on several datasets show the effectivenesses.

**Q2-3 Extent To Which Claims Are Supported By Evidence:**

3: Good: the main claims are supported by convincing evidence (in the form of adequate experimental evaluation, proofs, (pseudo-)code, references, assumptions).

**Q2-4 Reproducibility:**

3: Good: key resources (e.g. proofs, code, data) are available and key details (e.g. proofs, experimental setup) are sufficiently well-described for competent researchers to confidently reproduce the main results.

**Q3 Main Strengths:**

- The proposed method is efficient for removing false detections.

- The proposed method is well-motivated, based on the reasonable analysis.

**Q4 Main Weakness:**

None

**Q5 Detailed Comments To The Authors:**

In Sec. 3.1, regarding the definition of FD, $y\notin D$ should be $y\in D$?

**Q9 Complying With Reviewing Instructions:**

Yes

---

> ### Author Rebuttal · Authors · 2024-04-02
>
> We are grateful for the reviewer's insightful and clear feedback. Their observation regarding the discrepancy in Section 3.1 about the definition of false detections (FD) is especially appreciated. The accurate definition, as correctly noted by the reviewer, is indeed $y \in \mathcal{D}$, while $y^* \notin \mathcal{D}$. We have corrected this so that now $\mathrm{FD} = $ \{$y \in \mathcal{D} \mid y^*\notin \mathcal{D} \mid c_{y} \neq c_{y^*}$ or $\mathrm{IoU}(y^*, y) < \tau$\}. We remain committed to ensuring clarity and accuracy in all definitions and notations. We thank the reviewer once more for their valuable contribution to enhancing the quality of our work and their positive feedback.

---

### Official Review · Reviewer_3B2J · 2024-03-21

**Q2-1 Originality-Novelty:** 3
**Q2-2 Correctness-Technical Quality:** 3
**Q2-5 Clarity Of Writing:** 2

**Q1 Summary And Contributions:**

1. The authors propose a cost-sensitive framework for object detection tailored to user-defined budgets on the two types of errors, missing and false detections.
2. The authors derive minimum thresholding requirements to prevent performance degradation and define metrics to assess the applicability of uncertainty for failure recognition.
3. The authors automate and optimize the thresholding process to maximize the failure recognition rate w.r.t. the specified budget.

**Q2-3 Extent To Which Claims Are Supported By Evidence:**

3: Good: the main claims are supported by convincing evidence (in the form of adequate experimental evaluation, proofs, (pseudo-)code, references, assumptions).

**Q2-4 Reproducibility:**

3: Good: key resources (e.g. proofs, code, data) are available and key details (e.g. proofs, experimental setup) are sufficiently well-described for competent researchers to confidently reproduce the main results.

**Q3 Main Strengths:**

1. The authors have made a full research on the related work and summarized it in place. In the introduction and related work, the author clearly expounds the motivation of this study, and expounds the similarities and differences between this paper and other work, highlighting the research significance of this paper.
2. The authors have made a clear research and analysis on the correlation between the uncertainty-based thresholding and error detection.

**Q4 Main Weakness:**

Although the authors make a detailed analysis of the research content of this article,they don’t explain some details, which make some places difficult to understand. Moreover, there are many expressive problems (colloquialism and grammar) in the article, and the writing needs to be improved.

**Q5 Detailed Comments To The Authors:**

1.	The whole article focuses on the relationship between the uncertainty-based thresholding and error detection, and it is suggested that the author explain the uncertainty-based thresholding in detail.
2.	In Section 3.1, what do CDT, FDT, u(σ,δ)=1 and u(σ,δ)=0 respectively mean? What is the decision threshold? Could the authors can give a more detailed explanation?
3.	The language of the article is not refined enough, and the text description is not clear, which makes it difficult for people to understand, for example “The true category, i.e., the thresholding ground truth, of a detection y is defined by its classification result (cy) and its intersection over union (IoU) ...”.
4.	For the experimental part, the authors don’t compare with the existing methods, and suggested adding the latest method to compare with this method.

**Q9 Complying With Reviewing Instructions:**

Yes

---

> ### Author Rebuttal · Authors · 2024-04-02
>
> We thank the reviewer for their detailed and constructive feedback. We acknowledge the reviewer's concerns regarding the lack of clarity, which we diligently addressed. First, we elaborated on the definition of uncertainty-based thresholding and restructured Section 3.1 with more detailed explanations. We thereby enhanced the definition of $u(\sigma,\delta)$, with $u=1$ when the uncertainty $\sigma$ predicted by the model surpasses the uncertainty threshold $\delta$, signaling that the detection is subject to thresholding; and $u=0$ when the uncertainty falls below this threshold, implying that the detection is retained.
>
> In addition, we provided detailed context for the terms $\mathrm{CD_{T}}$, $\mathrm{FD_{T}}$, which represent the results of thresholding as mentioned in Section 3.1. Here, $\mathrm{CD_{T}}$ denotes the detections assigned as correct post-thresholding ($u=0$), while $\mathrm{FD_{T}}$ represents the detections assigned as false ($u=1$). Throughout, we have specifically rephrased complex sentences, such as the one mentioned by the reviewer; with the aim to further improve readability.
>
> In response to the valuable suggestion of comparing our work with existing methods, we wish to highlight that in Section 4 of our submitted paper, we present a comparison between our approach of optimally combined uncertainties for thresholding and the methods in related work that utilize unprocessed uncertainties as directly provided by the model. Furthermore, we wish to highlight the innovative aspect of our research that incorporates a budget-based cost-sensitivity in object detectors. To the best of our knowledge, this has not been addressed in prior work and therefore limits direct comparisons in this domain.
>
> We are fully committed to engaging with further feedback and strive to ensure our paper reflects the highest standards. We value the reviewer's insightful comments and suggestions, and are thankful for the chance to refine our work based on their expert guidance.

---

### Official Review · Reviewer_fizg · 2024-03-21

**Q2-1 Originality-Novelty:** 3
**Q2-2 Correctness-Technical Quality:** 3
**Q2-5 Clarity Of Writing:** 3

**Q1 Summary And Contributions:**

The paper presents an automated method for thresholding failures in probabilitic detectors (where there is a location and classification to be detected). The method is tested against KITTI and BDD100K datasets, and the false detection rate is significantly reduced, at only a slight increase in missing detections.

**Q2-3 Extent To Which Claims Are Supported By Evidence:**

3: Good: the main claims are supported by convincing evidence (in the form of adequate experimental evaluation, proofs, (pseudo-)code, references, assumptions).

**Q2-4 Reproducibility:**

3: Good: key resources (e.g. proofs, code, data) are available and key details (e.g. proofs, experimental setup) are sufficiently well-described for competent researchers to confidently reproduce the main results.

**Q3 Main Strengths:**

The introduction is strong, and the reader is left with a well defined problem awaiting to be solved. The results are also rigourous, testing against state of the art methods in well accessible datasets.

**Q4 Main Weakness:**

The description of the method (Section 3) lacks explanatory detail amongst the technical rigor making it hard to follow at times.

**Q5 Detailed Comments To The Authors:**

Overall the paper is great, though I had trouble following the detail in Section 3. In particular, in Section 3.2, some further explanation/motivation could be given for (1a). I also couldn't see a defintion for $i$ or $m$, which would help.

Perhaps for readability, you could consider breaking the subsections of Section 4 into paragraphs, rather than a wall of text that is easy to get lost in.

Otherwise, I think the paper is in a good shape, and I couldn't find any grammar/typos/minor mistakes)

**Q9 Complying With Reviewing Instructions:**

Yes

---

> ### Author Rebuttal · Authors · 2024-04-02
>
> We deeply appreciate the generally positive feedback and constructive suggestions to improve our paper. In response, we undertook a comprehensive revision of Sections 3 and 4, focusing on providing greater detail and improving readability. Specifically, in light of the reviewer's insightful feedback, we first extended the explanation surrounding Eq. (1a). Briefly summarized, we determine in Eq. (1a) the impact of detection removal on detector performance by evaluating changes in recall, precision, and the F1-score before and after thresholding. This analysis allows the definition of the requirements on the thresholding performance with regard to the detector, thereby directly linking thresholding efficacy to overall detector performance. The requirements for a given budget are then presented in Eqs. (1b) and (1c).
>
> Furthermore, we have clarified all necessary symbols and terminology, such as the proportion of remaining correct detections ($i$) and removed false detections ($m$) post-thresholding. This includes a refined description of our thresholding approach in Section 3. Acknowledging the reviewer's suggestion, we have also segmented Section 4 into shorter paragraphs, each dedicated to a single idea, thereby enhancing the narrative flow and readability of our paper.
>
> We hope that these revisions address the reviewer's concerns. Our objective remains to present our findings in a manner that is both accessible and thorough, without compromising the technical rigor of our work. We are keenly open to further feedback that may refine our work and extend our gratitude to the reviewer for their helpful suggestions.

---

### Meta-Review · Area_Chair_ZvAk · 2024-04-17

The paper presents a cost-sensitive and uncertainty-based method for object detection.  All reviewers are positive about this paper; it receives three reviews: 1 strong accept, 1 accept, and 1 weak accept.    They find the paper well written and the problem well-motivated.  The ideas are novel, and the experimental results are thorough and convincing.  The reviewers encourage the authors to improve the paper’s writing and presentation clarity. The authors rebuttal addresses some of the reviewers concerns.